# A Systematic Review of the Factors That Influence Teachers’ Occupational Wellbeing

**DOI:** 10.3390/ijerph20126070

**Published:** 2023-06-06

**Authors:** Joy C. Nwoko, Theophilus I. Emeto, Aduli E. O. Malau-Aduli, Bunmi S. Malau-Aduli

**Affiliations:** 1College of Medicine and Dentistry, James Cook University, Townsville, QLD 4811, Australia; 2College of Public Health, Medical and Veterinary Sciences, James Cook University, Townsville, QLD 4811, Australia; 3School of Environmental and Life Sciences, College of Engineering, Science and Environment, University of Newcastle, Newcastle, NSW 2308, Australia; 4School of Medicine and Public Health, College of Health, Medicine and Wellbeing, University of Newcastle, Newcastle, NSW 2308, Australia

**Keywords:** wellness, teacher wellbeing, stress and burnout, resilience, coping strategies, autonomy, student behaviour, classroom management, emotional competence, teacher self-efficacy

## Abstract

Teachers belong to a high-demand occupational group and experience work-related challenges and discretely diverse emotional turmoils of varying intensity while teaching and interacting with students. These experiences often result in high stress levels that contribute to burnout and, consequently, a breach of teachers’ occupational wellbeing. Promoting positive teacher wellbeing substantially influences teaching quality, with a flow-on effect on student wellbeing and academic development. This literature review utilised a framework to systematically explore the factors that impact the occupational wellbeing of kindergarten, primary, and secondary schoolteachers. Thirty-eight (38) studies from an initial 3766 peer-reviewed articles sourced from various databases (CINAHL, Emcare, PychINFO, Scopus, ERIC, and PsycARTICLES) were utilized for this systematic review. Four major factors were identified, including personal capabilities, socioemotional competence, personal responses to work conditions, and professional relationships. Findings highlight the importance of teachers’ occupational wellbeing in dealing with numerous challenges and competing demands, with the need for a high level of self-efficacy for instruction and behavioural management being critically significant. Teachers require adequate organisational support to successfully carry out their roles with stronger resilience and efficient job execution. Teachers also need to have social–emotional competence to be able to create a high-quality classroom environment and a conducive atmosphere that supports healthy teacher–student relationships, reduces stress and increases the occupational wellbeing of teachers. Collaborating with other relevant stakeholders such as parents, colleagues, and a school’s leadership team is critical for creating a positive work environment. A good workplace has the potential to contribute to teachers’ occupational wellbeing and provide a supportive platform for student learning and engagement. This review clearly points to the beneficial effects of prioritising teachers’ occupational wellbeing and its intentional inclusion in the professional development plan of practising teachers. Finally, while primary school teachers and secondary school teachers share many similarities in terms of the challenges they face, there are also some differences in how these challenges impact their wellbeing, and these warrant further investigation.

## 1. Introduction

The teaching profession can be incredibly rewarding and fulfilling [1,2], but teachers deal with many challenges and demands that can affect their occupational wellbeing and influence their ability to create a supportive learning environment for students [3,4]. Occupational wellbeing refers to the ability to achieve balance between work and leisure in a way that promotes health and a sense of personal satisfaction, which, for most people, is also financially rewarding and satisfying [5,6]. However, maintaining the occupational wellbeing of today’s working professionals is a real challenge [7]. Research has shown that teachers experience higher stress than many other professions [8,9]. Since teachers help to shape the future of students [10,11] and teacher wellbeing affects the classroom environment [12], occupational wellbeing should not be overlooked or taken for granted [11]. Teacher wellbeing has been defined as “an individual sense of personal professional fulfilment, satisfaction, purposefulness and happiness, constructed in a collaborative process with colleagues and students” [13].

The 21st century teacher’s role is increasingly challenging and complex [14], with teachers frequently reporting poorer mental health compared with workers in other professions [15,16]. The teaching profession exposes teachers to a variety of discrete emotions of different intensity [17,18,19,20,21]. The profession has been acknowledged as being very demanding, challenging, and stressful at all levels of teaching [3,22]. This is partly because governments across the world regularly implement reforms directed at transforming and improving education [23] with a magnitude and rapidity that often affect the occupational, emotional, and physical wellbeing of teachers [24]. The resultant pressures and challenges impact the quality of teaching, leading to decreased job satisfaction, increased emotional exhaustion, and decreased occupational wellbeing of teachers [25].

It is, therefore, not surprising that, globally, teachers are increasingly reporting elevated levels of occupational stress [26,27], exhaustion, anxiety, and burnout [28] as well as extremely demanding work intensification burdens [29]. This generates excessive workload and places undue pressure on teachers’ work–life balance, which in turn leads to a decline in morale and job satisfaction [29,30]. Jepson and Forrest (2006) [31] argued that teachers who experience stress in their work environments are more likely to feel a lower sense of occupational commitment. Primary school teachers have reported moderate-to-high levels of stress at work [3,26] and being more susceptible to high emotional exhaustion [32]. Additionally, teachers have been reported to work for 43 h per week on average [33]. Such excessive administrative work has been considered as a major source of stress, with higher levels of perceived stress amongst female teachers [34,35]. Globally, there are concerns that only a few people are considering teaching as a career option and many teachers exit the profession within a few years of starting their career [8]. Although there are differences in teacher attrition rate in different countries, the average rate of loss to the teaching profession is around 40–50% over the five years post-entry into the profession in many countries [36]. Brill and McCartney [37] noted that 33% of teachers in the United States leave their schools in the first three years, and 46% after five years.

Recent research has suggested that teachers’ stress and burnout is increasing at an alarming rate and, consequently, affecting overall satisfaction and professional engagement [38,39]. Teachers with high self-efficacy beliefs consider themselves well-equipped to cope with the stressful job demands and employ the use of available resources efficiently [40,41]. High levels of teacher self-efficacy result in increased positive emotions [42] and enhanced wellbeing [43,44,45,46]. In a Spanish study of 413 primary and secondary school teachers, Bermejo-Toro et al. [41] reported that teachers perceived self-efficacy and coping skills as critical to their wellbeing. On the other hand, low teacher efficacy often results in poor management of stress with the resultant exhibition of non-coping behaviours [47]. Low teacher efficacy has also been associated with higher levels of stress and burnout [48]. Consequently, low self-efficacy triggers increased work-related stress and depression [42,49].

There have been calls for actions to be taken to improve teacher wellbeing [39]. In their review of the literature on teacher wellbeing, McCallum and colleagues [8] reported that teacher wellbeing is important for the future of education because teachers are the most important contributors to student achievement, success, and satisfaction in the school environment. The authors emphasised the importance of focusing on teacher wellbeing to ensure creation of healthy learning environments [8]. The overarching research question for this systematic review was: What are the factors that impact teachers’ occupational wellbeing in the classroom? Specifically, this systematic literature review aimed to (1) investigate the factors that impact on teachers’ occupational wellbeing from kindergarten to secondary school educational levels, (2) identify existing knowledge gaps, and (3) make appropriate recommendations for future research.

## 2. Materials and Methods

This systematic review was conducted and reported in accordance with the Preferred Systematic Reviews and Meta-Analyses (PRISMA) Statement [50].

### 2.1. Inclusion and Exclusion Criteria

The inclusion criteria for this review were: (1) peer-reviewed articles written in English (2) conducted in the last 20 years (between 2002 and 2022), (3) related to the teaching profession from kindergarten to secondary level (4) focused on teacher occupational wellbeing within the school environment. Studies were excluded if they did not report the above characteristics or were literature reviews.

### 2.2. Search Strategy

Electronic databases including Medline, CINAHL, Emcare, PyschInfo, Scopus, Eric, and PsycARTICLES were searched from March 2022 to November 2022 for peer-reviewed articles that met the inclusion criteria. MeSH subject heading, subject heading terms, phrase searching, and truncation were used to aid searching. The Polyglot Search Translator [51], within the Systematic Review Accelerator software package (developed by Bond University, Gold Coast, Australia: https://bond.edu.au/iebh/systematic-review-accelerator-sra (accessed on 10 May 2023)), was used to properly align the search terms for each database. The reference list of the articles that met the inclusion criteria were hand-searched for additional relevant articles. Appendix A portrays the applied search terms.

### 2.3. Data Extraction and Synthesis

In this study, two authors (JCN and BMA) identified and independently screened titles and abstracts of retrieved articles. Articles that did not meet inclusion criteria were excluded while full text of potentially eligible articles were screened for inclusion. Disagreements were resolved in a consensus meeting.

A data extraction form was developed and used to collect relevant information from all the included studies. Descriptive data including author details, year of publication, study aims, design, location, and participant details were extracted from each of the selected studies. Each article was analysed for statements and concepts related to teachers’ work and wellbeing. Significant factors that impact teacher wellbeing, as either enablers or barriers, were identified and classified into four groups based on the Teacher Wellbeing Framework by McCallum et al. [8]. These included (1) teachers’ personal capabilities, which relate to resilience, authenticity, and self-efficacy, i.e., positive adaptations and self-judgments about their ability to positively influence student outcomes; (2) socioemotional competence, which relates to the identification, processing, and regulation of emotions; (3) personal responses to work conditions, which are characterised by emotional fatigue, disengagement, irritability, and apathy resulting from the work environment; and (4) professional relationships, which relate to student misbehaviour, issues with parents, support or lack thereof from management and leadership, and challenging situations that arise with colleagues. This framework was used to guide the review process and aid in better understanding of knowledge gaps.

### 2.4. Risk of Bias Assessment

The Quality Assessment Tool for Studies with Diverse Designs (QATSDD), which allows researchers to compare studies with different research designs [52], was used to critically assess the quality of the included articles. The QATSDD tool utilises 16 indicators for evaluation and each of the indicators is measured on a 4-point Likert scale that ranges from 0 to 3 (0 = not at all, 1 = very slightly, 2 = moderately, 3 = complete; n/a = not applicable). The obtained scores were summed up and expressed as a percentage of the maximum possible score to assess the quality of the included studies. The indicators included: (1) explicit theoretical framework; (2) statement of aims/objectives in main report; (3) clear description of research setting; (4) evidence of sample size considered in terms of analysis; (5) representative sample of target group of a reasonable size; (6) description of procedure for data collection; (7) rationale for choice of data collection tool(s); (8) detailed recruitment data; (9) statistical assessment of reliability and validity of measurement tool(s) (quantitative studies only); (10) fit between research question and method of data collection (quantitative studies only); (11) fit between research question and format and content of data collection tool (qualitative studies only); (12) fit between research question and method of analysis (quantitative studies only); (13) good justification for analytic method selected; (14) assessment of reliability of analytic process (qualitative studies only); (15) evidence of user involvement in design; (16) strengths and limitations critically discussed. In order to aid interpretation, the scores obtained for all indicators within each article were summed up and converted into percentages. Articles with scores over 80% were classified as high quality, scores from 50% to 80% were classified as medium quality, while articles below 50% were classified as low quality.

## 3. Results

### 3.1. Article Selection

The initial search yielded 3766 peer-reviewed articles that were imported into Endnote. After duplicates were removed and hand-searched articles (20 in number) were added, 3566 articles were found to be potentially relevant to the research topic. After screening the potentially relevant articles, 247 articles were retrieved. Based on abstracts and titles screening, the number was reduced to 221. Full texts were explored for eligibility and 38 studies were included for the review. The PRISMA flowchart of the literature search is presented in Figure 1.

### 3.2. Characteristics of the Reviewed Articles

Table 1 presents the summary characteristics of the included studies. The studies originated from 21 different countries: Australia (*n* = 5), Belgium (*n* = 1), Canada (*n* = 2), China (*n* = 2), Croatia (*n* = 4), Czech Republic (*n* = 1), Finland (*n* = 3), Germany (*n* = 2), Ghana (*n* = 1), Ireland (*n* = 1), Israel (*n* = 1), Italy (*n* = 1), Malaysia (*n* = 1), the Netherlands (*n* = 1), Poland (*n* = 1), Romania (*n* = 1), Spain (*n* = 1), the United Kingdom (*n* = 3), and the United States (*n* = 3). There were also combined studies between countries: Italy and Switzerland (*n* = 3), Syria and Germany (*n* = 1). Total number of participants was 22,675 with a range from 5 to 3010. Participants’ ages ranged from 20 to over 60 years and they were predominantly females (*n* = 17,359). In relation to school type, there were 750 participants from pre-primary (kindergarten); 10,736 from primary; 7827 from secondary; 97 from special education schools; and 413 from vocational schools. The school type of 2852 participants was not stated. Of the 38 reviewed articles, 20 of them focused on primary school teachers.

The research designs included quantitative (*n* = 31), mixed-method (*n* = 1), and qualitative (*n* = 6) variants. Diverse validated tools/scales were used in the quantitative studies to measure factors that affect teachers’ occupational wellbeing, while semi-structured interview questions were used for qualitative studies. Each of the quantitative studies evaluated multiple factors and discussed them as such. Out of the 38 studies reviewed, 31 used measurement instruments, and the three most used instruments were the Maslach Burnout Inventory—Educators Survey (MBI-ES), Teacher Self-Efficacy Scale, and the Work Engagement Scale. Moreover, 17 studies used frameworks, and the most frequently used framework was The Job Demands–Resources Model (JD–R Model).

### 3.3. Teachers’ Perceptions of Factors That Impact Their Occupational Wellbeing

As shown in Table 2, four overarching factors that affect teachers’ occupational wellbeing and either positively or negatively impact their professional aptitude were identified: (1) personal teacher capabilities; (2) socioemotional intelligence; (3) personal responses to workload and work conditions; and (4) professional relationships. Elements of personal capabilities included resilience, self-efficacy, autonomy, and coping strategies. Emotional intelligence components included emotional competence, training opportunities, and other supportive work relationships, while personal responses to workload and work conditions included burnout, fatigue, exhaustion, stress, unrealistic expectations, stress, bureaucracy, and exclusion from decisions. Other relational factors that were perceived to influence teachers’ wellbeing included student misbehaviour, misunderstandings with parents and colleagues, and perceived lack of support from the school management system.

#### 3.3.1. Personal Teacher Capabilities

Teachers’ perceptions of self-efficacy, resilience, authenticity, and coping strategies were discussed in 23 studies [7,23,41,42,43,44,45,46,48,53,54,55,56,57,58,60,61,63,64,67,70,73,74]. Across all teaching levels, high self-efficacy resulted in quality teaching, positive emotions [42], and low burnout amongst primary school teachers [70]. It served as a protective resource for high school teachers [48] and enhanced occupational wellbeing for all teachers [41,43,44,45,46,55,58,64,73]. Positive emotions were highly associated with self-efficacy and wellbeing, as teachers with high self-efficacy were able to provide higher quality instruction and had greater power in promoting students’ motivational, affective, and cognitive outcomes [44,52]. This implies that teacher self-efficacy enhanced positive emotion. On the other hand, negative emotions led to strong intentions to quit the job amongst primary and secondary school teachers [7]. High resilience led to lower levels of psychopathological symptoms and burnout in primary and secondary school teachers [54], and improved leadership, communication, and workplace wellbeing for secondary school teachers [57]. Adaptive coping strategies [7,67], social support [41,56], exercise, and meditation [60] were used to maintain wellbeing. Some teachers used depersonalisation as a form of defensive coping mechanism [61]. Furthermore, low self-efficacy preceded burnout, particularly among younger teachers [48].

#### 3.3.2. Socioemotional Competence

Teachers’ perception of their emotional competence, supportive relationships, and training opportunities were discussed in 17 studies [7,14,33,38,42,44,45,46,53,55,56,58,64,68,69,73,74].

Positive teacher emotions enhanced self-efficacy [42,44] and wellbeing [53], while negative emotions led to strong quitting intentions [7]. Difficulty in regulating emotions resulted in stress [33] but could be improved through training [46]. Teachers reported decreased socioemotional competence during the COVID-19 pandemic [74]. During the pandemic, teachers felt increasingly uncertain, overwhelmed, and stressed, which was due to having to combine work and family life in isolation and with limited support [74]. Nonetheless, a positive teacher–student relationship fostered occupational wellbeing [45,55,58,64,73,74]. Teachers preferred interacting with students directly [58]. Teacher wellbeing was affected by stressful experiences of student misbehaviour [14]. When teachers struggled with regulating their emotions, it led to greater stress [33,38,56]. Social, emotional and academic functioning of special education students improved because of teachers’ training [46]. Mindfulness training did not help with socioemotional competence [68]. Difficulty in regulating teacher emotions resulted in stress for some primary and secondary school students [33] but could be improved through training [46]. Mindfulness training improved primary and secondary school teachers’ competence in behavioural management [68], while being positive led to more cooperative behaviour and better learning outcomes [69].

#### 3.3.3. Personal Responses to Work Conditions

Teachers’ personal responses to work conditions were discussed in 25 studies [3,14,22,23,33,38,43,45,54,56,58,59,60,61,62,64,65,66,67,69,70,71,72,73,74]. Teachers at all levels reported high workload [22,23,43,45,58,60,71,72] and classroom demands [45,60,71], which resulted in stress, burnout [22,72], and mental health/wellbeing problems [60]. Reduced workload led to a decrease in psychosomatic complaints and fatigue and fostered better health outcomes [58]. Many teachers found their job very stressful [23,33,64] due to lack of support [23] and classroom challenges [3]. Teachers also reported stress due to anxiety, depression, and unrealistic expectations during the COVID-19 pandemic due to remote teaching [45,59,74]. For example, school leadership wanted teachers to provide meaningful lessons and assessment/feedback virtually to students within normal working hours with no overtime granted. Teachers felt the expectation was unrealistic because administrators wanted more with limited resources [74].

Autonomy was noted as being critical to occupational wellbeing because it increased teacher motivation and job satisfaction, particularly when combined with reflective practice [43,60,63]. Increased bureaucratic processes and unclear role expectations decreased autonomy for primary and secondary school teachers [23]. Despite high self-efficacy, where role expectations were unclear, teachers reported low levels of autonomy, felt emotionally exhausted, undervalued, unappreciated, deficient, and inadequate [23,45]. Additionally, during the COVID-19 pandemic, teacher experiences indicated reduced autonomy and flexibility, which impacted their occupational wellbeing [74].

Teacher emotional exhaustion was as a result of teaching and intense interactions with students [3,42,54], emotional intensity and dissatisfaction with levels of support [38,56], low occupational wellbeing [73], work intensification [62], and inability to effectively manage professional and family roles [69]. There was less emotional stress and exhaustion as a result of training [46]. High physical exertion, job demands, [22,61], and mental health symptoms [66] resulted in teacher burnout. To prevent burnout, teachers needed to be optimistic and receive social support [65,70]. Work environment affected perceived stress, physical and mental wellbeing, and job satisfaction [14,33,66,74]. High job demands and stress led to high somatic complaints or ill health [22,61] while reduced job demands resulted in less fatigue and psychosomatic complaints [58].

Teacher job satisfaction increased and was related to wellbeing, motivation [55,64,69], lower burnout [70], and low somatic complaint [67]. Teachers reported high job satisfaction as a result of availability of resources [64,69], positive emotions [14], social support [64,69,70], and face-to-face class interactions [58]. Less job satisfaction and motivation were mainly due to anxiety and depressive symptoms [66], and nonteaching-related workload [62]. Work motivation played an important role in teachers’ wellbeing [63] and was related to job satisfaction and commitment [55]. The availability of job resources fostered wellbeing [71], mental health [69], and increased commitment [22]. A decrease in job resources resulted in a decrease in job satisfaction at all teacher levels [58].

Overall, primary classroom teachers were the most concerned about the issue of competing demands on their time [60]. Kindergarten teachers struggled to maintain a balance between personal and professional responsibilities [43]. Primary school teachers, especially, found their jobs extremely stressful [31,33] compared to secondary school teachers, who reported moderate levels of stress [31]. Coaching reduced stress for secondary school teachers [57]. Higher stress from workload led to demotivation [64]. In an Australian study of 749 teachers, primary and secondary school teachers indicated burnout experiences [33]. Male teachers experienced more and stronger emotional burnout [70] than female teachers [38,56]. Female teachers experienced stronger physical burnout [70]. Burnout led to negative emotions, deterioration in mental health, and psychopathological symptoms [54]. Generally, teachers, irrespective of their country, were similar in their perception of factors that affected their burnout, stress levels and personal accomplishment. However, there were significant differences in relation to satisfaction with professional support from colleagues and the organisation. For example, Belgian teachers reported higher physical exertion, job demands, and somatic complaints, but lower job control, social support, and personal accomplishment than teachers from other European countries [61]. However, older teachers reported higher somatic complaints than their younger colleagues [67].

#### 3.3.4. Professional Relationships

Twenty-one (21) studies [3,7,14,22,33,38,43,46,48,53,54,55,58,59,61,62,66,68,69,71,73] addressed organisational commitment, job resources, and students’ behaviour. The studies argued that it is critical for organisations to implement working policies for all teachers in all school types [38,43] and to provide intervention training workshops to teachers. Training programmes on how to use reappraisal strategy [53], regulate emotion [59], improve teaching skills [48], connect well with students [14], improve coping strategies [7], manage workload and reduce stress [33], use redesigned strategies [46], and handling of digital tools [58] were needed. The studies also suggested that organisational strategies and support [38,43] were required for teachers, especially teachers with family care responsibilities [58]. This was necessary to reduce burnout and dissatisfaction with the profession [62], meet teachers’ mental health needs [66], and enhance all teachers’ occupational [22,46,54,73] and psychological wellbeing [55,69]. Job resources increased teacher wellbeing [71], commitment [22], and mental health [69]. Fewer job resources resulted to a decrease in job satisfaction and more challenges [58]. However, insufficient emotional resources led to depersonalization [61]. Managing disruptive student behaviour was stressful for primary school teachers [3] and affected their wellbeing [14]. Positive moods resulted in more cooperative behaviours from students in primary and secondary schools [69]. Training improved primary and secondary school teachers’ competence in behaviour management [46,68]. Comparison of responses by country indicated that lack of support from colleagues resulted in burnout, in some settings, while organisational identification had a stronger impact on burnout among others [65].

### 3.4. Quality Appraisal of the Reviewed Journal Articles

As portrayed in Table 3, the QATSDD assessment indicated that about 89.5% (*n* = 34) of the included studies were of medium quality [3,7,14,22,23,31,33,38,39,41,43,44,48,53,54,55,56,57,58,59,60,61,62,63,65,66,67,68,69,70,71,72,73,74] and none (*n* = 0) were of low quality. Individual scores ranged from 52.4% to 81%. The top-quality studies were four in number. Three of the high-quality studies were quantitative while one [45] was a mixed-methods study. They were judged to be explicit in their methodology while most of the medium-quality studies were quantitative. Some of the weaknesses identified from the medium-quality studies included: lack of explicit theoretical framework, inadequate sample sizes, poor justification for analytical method selected, inadequate evidence of user involvement and poor/absence of critical discussion of strengths and limitations of study. The four top-quality studies [42,45,46,64] utilised theoretical frameworks to substantiate their research and provide in-depth understanding of the phenomenon [75].

## 4. Discussion

Teachers at all levels face enormous challenges daily in a profession that is considered highly stressful when compared to other professions [22,41,74]. This systematic review of 38 studies explored and synthesised the perceptions of kindergarten to secondary school teachers about the factors that impact their occupational wellbeing. Four categories of factors were found to be positively or negatively associated with teachers’ occupational wellbeing.

The reviewed studies suggest that personal capabilities such as self-efficacy, resilience, coping strategies, and autonomy are critical to the occupational wellbeing of teachers [41,64]. Teachers with high self-efficacy were found to be associated with quality teaching, positive emotions, and low burnout levels [76,77], which corroborates the findings of Nuallaong [49]. This means that the higher a teacher’s self-efficacy, the lower the burnout risk and the better the teacher is in enhancing students’ learning [78]. Similarly, high resilience was found to lead to lower levels of psychopathological symptoms and burnout among teachers. Adaptive coping strategies, social support, exercise, and meditation were also found to be helpful in maintaining teacher wellbeing. The importance of autonomy in promoting teacher motivation and job satisfaction was also highlighted [45,55,73]. However, bureaucratic processes and unclear role expectations were noted to decrease autonomy, which negatively impacted teacher wellbeing [23]. This highlights the need for clarity and transparency in role expectations and bureaucratic processes to promote autonomy among teachers. The COVID-19 pandemic also impacted teacher autonomy and flexibility, leading to reduced occupational wellbeing [73]. This finding highlights the need for supportive measures to help teachers navigate challenging situations and maintain their wellbeing.

Teachers’ socioemotional competencies, including their emotional competence, supportive relationships, and training opportunities, are also critical components of their occupational wellbeing. The review highlights the importance of positive teacher–student relationships in fostering occupational wellbeing. Teachers who interacted positively with their students reported better occupational wellbeing. However, negative experiences of student misbehaviour were associated with decreased teacher wellbeing. For example, it was reported that primary school teachers’ active response to emotional job demands led to improvement in their relationships with students [44]. On the other hand, if teachers experienced difficulties in regulating and controlling their emotions while working with students, it led to stress [33]. This may be due to the fact that being positive leads to more cooperative behaviour from students and better learning outcomes [69,79]. Additionally, building teacher–student relationships assisted the teacher in better understanding a student’s challenging behaviour, thus enabling the teacher to show greater concern and empathy [80]. This emphasises that teachers’ ability to regulate their emotions plays a crucial role in how they respond to student misbehaviour, highlighting the importance of emotional competence in promoting positive teacher–student relationships. Furthermore, the review emphasizes the importance of training opportunities for teachers to improve their socioemotional competencies. For instance, social, emotional, and academic functioning of special education students improved as a result of teachers’ training [46]. Mindfulness training was found to improve primary and secondary school teachers’ competence in behavioural management, while being positive led to more cooperative behaviour and better learning outcomes [68].

This review also revealed that teachers face numerous challenges in their work environment, including high workload, stress, burnout, exhaustion, and fatigue. Teachers at all levels reported high workloads [22,23,43,45,58,60,71,72]. The high workload and classroom demand were found to result in stress, burnout, and mental health problems. Teachers also experienced stress due to anxiety, depression, and unrealistic expectations regarding remote teaching. Emotional exhaustion was found to be a result of teaching and intense interactions with students, emotional intensity, dissatisfaction with levels of support, low occupational wellbeing, work intensification, and inability to effectively manage professional and family roles. The work environment affected the perceived stress, physical and mental wellbeing, and job satisfaction of teachers [81]. Primary classroom teachers were concerned about competing demands on their time. Availability of job resources fostered wellbeing, mental health, and increased commitment. Decrease in job resources resulted in a decrease in job satisfaction at all teacher levels. Primary school teachers found their jobs extremely stressful compared to secondary school teachers, who reported moderate levels of stress. Coaching reduced stress for secondary school teachers. Higher stress from workload led to demotivation.

To prevent burnout, teachers need to be optimistic, receive social support, and have high job satisfaction. The availability of resources, positive emotions, social support, and face-to-face class interactions were found to increase job satisfaction. Teachers who reported high job satisfaction also had high wellbeing and motivation. On the other hand, anxiety and depressive symptoms and nonteaching-related workload were found to decrease job satisfaction and motivation. Kindergarten teachers struggled to maintain a balance between personal and professional responsibilities [43]. Male teachers experienced more and stronger emotional burnout [38,56], while female teachers were found to experience stronger physical burnout [70]. More experienced primary and special education teachers had lower levels of burnout symptoms [72]. Forlin [82] had earlier suggested that years of service and participation in training programmes reduced stress for teachers. The review findings suggest that systematic support, in the form of intervention programmes, resources, workshops, implementation of work policies, and good relationship with management, is required as a means of improving mental health and wellbeing [58,64,73].

Furthermore, teachers’ wellbeing and job satisfaction are significantly impacted by their professional relationships with students and colleagues. Positive moods and behaviours were found to result in more cooperative student behaviour in primary and secondary schools. The reviewed studies suggested that training programmes focusing on various aspects of teaching, such as regulating emotions, improving teaching skills, and connecting well with students, can improve teachers’ wellbeing, job satisfaction, and commitment to their profession. Organisational support, including policies and intervention training workshops, was also deemed crucial for reducing burnout and dissatisfaction with the profession. This would be most beneficial to kindergarten and primary school teachers, who rely more on social support than others [65]. Moreover, job resources such as emotional support, adequate workload management, and coping strategies were found to positively impact teacher wellbeing, commitment, and mental health [66]. Support from the leadership has also been shown to reduce the risk of burnout and increase job satisfaction in teachers [83]. Similar findings from the literature consider social support a major job resource for teachers [41], having a significant impact on wellbeing [84,85]. Therefore, organisations should provide teachers with sufficient resources to enhance their wellbeing and mental health. Additionally, the studies indicated that disruptive student behaviour was a significant source of stress for teachers, particularly in primary schools [3,14]. Therefore, training in behaviour management and support from colleagues and school leaders may be necessary to help teachers manage such situations effectively.

### 4.1. Implications for Practice

This systematic review unequivocally portrays the teaching profession as being highly stressful. The implication for teaching practice is that a strengthening of organisational support—with professional workshops, intervention and training programmes to enhance teaching skills, occupational wellbeing, and mental health of teachers—is necessary [45,53,56]. There is also the need for a review of teaching workload aimed at stress reduction intervention programmes. Enhancement of teacher socioemotional competence is required, as is the provision of systematic support for the vulnerable. Collaborating with more experienced teachers as a shared responsibility is necessary, and so is the development of coping skills that might help reduce stress [14,39,60].

Based on the findings of the reviewed studies, there are some differences in the well-being of primary school teachers and secondary school teachers. Regarding personal capabilities, primary school teachers may require more energy and patience to handle younger children, which can be exhausting and require a high degree of emotional labour, while secondary school teachers may need more content knowledge and teaching skills to teach their specific subjects [42,54]. This suggests that the personal capabilities needed for each group of teachers may differ. In terms of socioemotional competence, both primary and secondary school teachers need to be skilled in managing their emotions and connecting well with students. However, primary school teachers may face more challenges in managing disruptive behaviour from young children, which can negatively impact their wellbeing [3,14]. On the other hand, secondary school teachers may need to deal with more complex social dynamics and interpersonal conflicts among students. When it comes to personal response to work conditions, primary school teachers may experience higher levels of workload and stress due to the constant need for attention and supervision of young children [74]. In contrast, secondary school teachers may have more autonomy in managing their workload but may also experience pressure to maintain high academic standards and prepare students for exams. Finally, in terms of professional relationships, both primary and secondary school teachers benefit from organisational support and resources, but primary school teachers may require more support due to the nature of their work with young children [65]. Additionally, secondary school teachers may face more pressure to maintain positive relationships with students and parents while also balancing their teaching responsibilities. Furthermore, teachers’ occupational wellbeing can be influenced by differences between countries and continents. Generally, there were significant differences in relation to satisfaction with professional support from colleagues and the organisation. It is important to note that individual countries within and across continents can have variations in their educational systems and teacher-related factors. Additionally, education landscapes and policies are dynamic and always evolving, which makes comparisons of teacher wellbeing factors between countries and continents quite challenging.

### 4.2. Future Research Considerations

Overall, while there are some similarities in the wellbeing challenges faced by primary and secondary school teachers, there are also some notable differences that can impact their wellbeing in unique ways. This highlights the need for targeted interventions and support for each group of teachers. Future research should particularly target primary school teachers whose occupational wellbeing seems to be more negatively impacted due the nature of their job and requirement for more social support. There is the need for further and comprehensive investigation into primary school teachers’ wellbeing using relevant frameworks to better understand their occupational wellbeing and the underlying associated variables within classroom settings, particularly in the areas of teacher workload, related burnout, and stress, as well as the benefits of socioemotional competence in creating a good quality classroom environment. Teacher wellbeing is also influenced by other contextual characteristics, such as school ownership (public or private), teacher career stage and subject specialisation. However, information on these variables was not provided in most of the articles reviewed in this study, pointing to the need for further research in this area.

### 4.3. Strengths and Limitations

Despite the carefully concerted effort to systematically search the literature, some studies might have been overlooked due to the inclusion and exclusion criteria used. The review was limited to only peer-reviewed papers written in English and published in the last 20 years. There is a likelihood that other relevant articles may have been missed. However, some of the strengths of this systematic review include an increased understanding of teachers’ occupational wellbeing and identified knowledge gaps in teacher training and support mechanisms required for managing the 21st century classroom and maintaining occupational wellbeing.

## 5. Conclusions

This systematic literature review explored the factors that impact teacher occupational wellbeing. It was evident that teaching as a profession is very stressful and emotionally exhausting, potentially leading to burnout and a negative impact on teachers’ occupational wellbeing. The current global political and educational agendas seem to prioritise child safety and student wellbeing, but teacher wellbeing and supportive school environments within educational settings should also be prioritised.

Effective management of classroom-related stressors using coping strategies, meaningful collaboration, and supportive colleagues are of immense assistance to teachers. Administrators within school settings are in ideal positions to support teacher occupational wellbeing. Support from experienced colleagues can help teachers to work effectively in improving student learning and wellbeing and to remain in the profession. When schools take steps to ensure the wellbeing and mental health of teachers in the classroom, teachers thrive and demonstrate resilience in the face of challenges that come with the teaching profession. In conclusion, this systematic review shows that teacher wellbeing is influenced by many factors including the school environment, work-related stress, and student–teacher relationships. The findings from this study also clearly indicated that primary school teachers were the most stressed; hence, future research should explore ways, means, and changes to their workload and stress, and ultimately seek ways of improving primary school teacher wellbeing.

## Figures and Tables

**Figure 1 ijerph-20-06070-f001:**
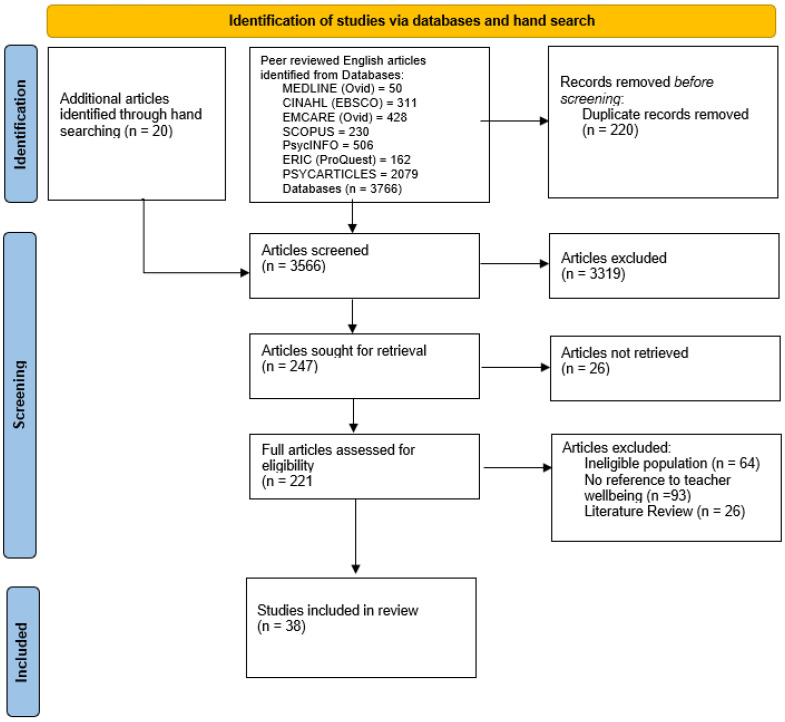
PRISMA flowchart of the systematic literature search.

**Table 1 ijerph-20-06070-t001:** Characteristics of the reviewed studies.

Author, Year, Title	Country of Study and Sample Size	Study Design	Participants and Sample Size	Instruments	Framework	Main Findings
Chaudhuri et al. (2021) [3]. Teachers’ focus of attention in first-grade classrooms: Exploring teachers experiencing less and more stress using mobile eye tracking.	Finland: *n* = 53; Female = 94%; Mean age = 44.6 years; Mean teaching experience = 16.07 years.	Quantitative	Primary Grade 1 teachers.	The Bergen Burnout Inventory (Finnish short version)	None	Focus and attention varied between more and less stressed teachers. Student behaviour management was stressful for teachers.
Wang et al. (2022) [7]. Coping profiles among teachers: Implications for emotions, job satisfaction, burnout, and quitting intentions	Canada: *n* = 947; Female = 82.3%; Mean age = 42.29 years; Mean teaching experience = 15.16 years	Quantitative	Primary (44.8%); Secondary (45.3%).	Coping Strategies Inventory; Social Desirability Scale; teachers’ psychological wellbeing	None	Teachers exhibited distinct and diverse patterns of coping strategies that varied in individual strength
Sidek et al. (2020) [14]. Student misbehaviour in classrooms at secondary schools and the relationship with teacher job well-being	Malaysia: *n* = 460	Quantitative	Secondary	Teacher Job Satisfaction Questionnaire (TJSQ); Emotional Exhaustion and Job Satisfaction Scale	None	Stress from student misbehaviour in the classroom and the wellbeing of teachers were at a moderately low level. Student misbehaviour was significantly associated with teacher wellbeing, which was positively related to job satisfaction but not associated with emotional exhaustion.
Hakanen et al. (2006) [22]. Burnout and work engagement among teachers.	Finland: *n* = 2038; Female = 79%; Mean teaching experience = 13.5 years.	Quantitative	Elementary (41.4%); Secondary (38%); Vocational (10.6%).	Maslach Burnout Inventory—General Scale (MBI-GS); Self-Rated Health Utrecht Work Engagement Scale (UWES); Healthy Organization Questionnaire (HOB)	Job Demands–Resources Model (JD–R)	Job demands and poor resources were directly associated with burnout. To prevent burn out and attrition, there was a need to enhance job resources in order to increase teachers’ commitment to the job.
Skinner et al. (2021) [23]. Managerialism and teacher professional identity: impact on well-being among teachers in the UK.	United Kingdom: *n* = 39; Female = 64%;	Qualitative	Primary and secondary school teachers and leaders.	Interview	None	The tension between teachers’ and managerial views of being a teacher impacted teachers’ professional identity, which led to mental health and emotional wellbeing challenges.
Jepson and Forrest (2006) [31]. Individual contributory factors in teacher stress: The role of achievement striving and occupational commitment	United Kingdom: *n* = 95; Female = 68%; Mean teaching experience = 12.3 years.	Quantitative	Primary (68%); Secondary (32%).	Perceived Stress Scale; Type A Behaviour Scale; Teacher Achievement Striving Scale (TASS); Teacher Occupational Commitment Scale (TOCS)	None	Individual contributory factors were significant for the prediction and understanding of teachers’ occupational stress. Primary school teachers reported the highest levels of perceived stress than secondary school teachers.
Carroll et al. (2021) [33]. Teacher stress and burnout in Australia: examining the role of intrapersonal and environmental factors.	Australia: *n* = 749; Female = 83%; Mean age = 44.27 years; Mean Teaching Experience = 17.66 years.	Quantitative	Primary (36%); Secondary (48%); Administrators (13%); Others = (3%).	Perceived Stress Scale (PSS); Teachers’ Perceived Context Scale (TPCS); the Difficulties in Emotion Regulation Scale (DERS); the Comprehensive Inventory of Thriving (CIT).	None	More than half of the teachers reported being very or extremely stressed and were considering leaving the profession. Early career teachers, primary teachers, and teachers working in rural and remote areas had the highest stress and burnout levels.
Stapleton et al. (2020) [38]. The effect of teachers’ emotional intensity and social support on burnout syndrome. A comparison between Italy and Switzerland.	Italy and Switzerland: *n* = 275; Italian = 140; Female = 82%; Swiss = 135; Female = 87%. Age = 25–60 years; Mean teaching experience = 12 years.	Quantitative	Primary	The Maslach Burnout Inventory—Educators Survey (MBI-ES); The Emotional Competence Questionnaire (ECQ); Social Support Questionnaire	None	Emotional intensity was significantly associated with emotional exhaustion and personal accomplishment dimensions of burnout in both Italy and Swiss teachers. Dissatisfaction with support received affected all measures of teacher burnout. However, the two groups differed significantly in satisfaction with support received.
Turner et al. (2022) [39]. Teacher wellbeing and social support: a phenomenological study	Australia: *n* = 5; Female = 100%; Years of teaching experience = 2–15 years.	Qualitative	Primary	Interview	PERMA wellbeing approach	The importance of understanding how teachers provided social support and the effects of socially supportive behaviours in schools on teacher wellbeing was evident. By providing social support at work, teachers were also supporting their own wellbeing.
Bermejo-Toro et al. (2016) [41]. Towards a model of teacher well-being: personal and job resources involved in teacher burnout and engagement	Spain: *n* = 413; Female = 71.6%; Mean age = 40.5 years.	Quantitative	Primary (51.4%); Secondary (48.6%).	Maslach Burnout Inventory—General Survey (MBI-GS) (Spanish version); Multidimensional Support Scale; Utrecht Work Engagement Scale (Spanish version)	The Job Demands–Resources Model (JD–R)	Teacher self-efficacy and appropriate positive coping styles elevated teacher wellbeing at work.
Burić et al. (2020) [42]. Teachers’ emotions and self-efficacy: A test of reciprocal relations	Croatia: *n* = 3010; Female = 82%; Mean age = 41.53 years; Mean teaching experience = 14.87 years.	Quantitative	Elementary (*29%*); Middle school (*35%*); Secondary school (*31%*).	Teacher Emotion Questionnaire (TEQ); Teacher Self-Efficacy Scale (TSES).	Structure model	Teachers with moderate-to-high levels self-efficacy beliefs reported higher levels of positive emotions, while those with higher levels of negative emotions reported lower levels of self-efficacy beliefs.
Kilgallon et al. (2008) [43]. Early childhood teachers’ sustainment in the classroom.	Australia: *n* = 63; Female = 97%.	Qualitative	Early childhood	Interview	None	Access to support and positive educational settings, attitudes, beliefs, and pedagogical practices contributed to early childhood teachers’ sustenance, as did their ability to maintain personal wellbeing and work–life balance.
Huang et al. (2019) [44]. Job characteristics and teacher well-being: the mediation of teacher self-monitoring and teacher self-efficacy.	China: *n* = 1115; Female = 76.1%; Mean Teaching experience = 20 years.	Quantitative	Primary	The emotional job demands of teaching; teachers’ perceived trust in colleagues; Self-Monitoring Scale; Self-Efficacy scale; Teacher Wellbeing Scale	Job Demands–Resources Model (JD–R)	Trust in colleagues helped enhance teacher self-efficacy and wellbeing. The rewarding side of emotional job demands laid in the enhancement of teacher self-efficacy.
Chan et al. (2021) [45]. Elementary school teacher well-being and supportive measures amid COVID-19: An exploratory study.	United States: *n* = 151; Female = 80.8%; Teaching experience = 2–46 years.	Mixed Methods	Primary	Maslach Burnout Inventory; Teaching Autonomy Scale; Role Ambiguity and Task Stress Subscales.	Job Demands–Resources Model (JD–R)	There was a critical need to provide more attention and resources to support teachers’ psychological health by strengthening autonomy, emotional support, and teaching efficacy.
Kennedy et al. (2021) [46]. Exploring the impact of incredible years teacher classroom management training on teacher psychological outcomes.	Ireland: *n* = 368; Mean teaching experience = 9 years.	Quantitative	Primary (76.6%); Special education (19.3%).	Teacher Sense of Efficacy Scale—Long Version (TSES-L); Everyday Feelings Questionnaire (EFQ); Maslach Burnout Inventory—Educators Survey (MBI-ES)	Prosocial Classroom Model	Training in classroom management led to positive psychological outcomes. Teacher wellbeing was critical for promoting positive mental health, social–emotional development, and student wellbeing.
Schwarzer and Hallum (2008) [48]. Perceived Teacher Self-Efficacy as a Predictor of Job Stress and Burnout: Mediation Analyses	Syria and Germany: *n* = 1203; Age = >21years; Syrian = 608 Female = 85%, German = 595 Female = 54%.	Quantitative	Syrian and German teachers	Teacher Self-Efficacy scale; General Self-Efficacy; Job Stress Scale; the Maslach Burnout Inventory (MBI)	None	Low self-efficacy preceded burnout, particularly in young teachers and others that had low general self-efficacy.
Burić, et al. (2021) [53]. Emotional labor profiles among teachers: Associations with positive affective, motivational, and well-being factors.	Croatia: *n* = 2002; Female = 83%; Mean Age = 42.34 years; Mean teaching experience = 15.95 years	Quantitative	Elementary (29%); Middle school (*36%*); Secondary 26.5%); Not stated = (*8.5%*).	Utrecht Work Engagement Scale The Job Satisfaction Scale; Teacher Self- Efficacy Scale; Emotional Labour Scale.	None	Teachers who mostly relied on deep acting had the most adaptive patterns of positive outcomes, whereas teachers who reported higher levels of hiding their feelings, regardless of the level of deep acting, exhibited lower desirable levels of positive effects, self-efficacy, work engagement, and job satisfaction.
Burić, et al. (2019) [54]. Understanding teacher well-being: A cross-lagged analysis of burnout, negative student-related emotions, psychopathological symptoms, and resilience.	Croatia: *n* = 941; Female = 83%; Mean age = 42.75 years; Mean teaching experience = 15.28 years.	Quantitative	Primary (*57.8%*); Secondary (*42.2%*).	Oldenburg Burnout Inventory; Psychopathological Symptoms by Symptom Inventory (BSI); Resilience Scale (BRS)	None)	Teachers with higher burnout levels had higher levels of negative emotions towards their students and more psychopathological symptoms. Higher levels of resilience led to lower levels of negative emotions, burnout, and psychopathological symptoms
Collie et al. (2016) [55]. Teachers’ psychological functioning in the workplace: Exploring the roles of contextual beliefs, need satisfaction, and personal characteristics	Canada: *n* = 485; Female = 76%; Mean age = 44 years; Mean teaching experience = 15 years.	Quantitative	Primary (50%); Middle (11%); Secondary (32%); More than one level (7%).	The Motivation at Work Scale; the Flourishing Scale; the Work-Related Basic Need Satisfaction Scale; Organizational Commitment.	Self-determination theory and teacher-hypothesized model.	Teachers’ psychological wellbeing at work was influenced by contextual beliefs, personal characteristics, and satisfaction. When the teachers’ work environment was autonomous and supportive, they felt that their basic psychological needs were being met.
Fiorilli et al. (2015) [56]. The effect of teachers’ emotional intensity and social support on burnout syndrome. A comparison between Italy and Switzerland.	Italy and Switzerland: *n* = 275; Italian = 140 Female = 82%): Mean teaching experience = 12 years. Swiss = 135 Female = 87%).	Quantitative	Primary:	The Maslach Burnout Inventory—Educators Survey (MBI-ES); Emotional Competence Questionnaire (ECQ)	None	Teachers were similar in their perception of factors that affected their burnout syndrome, personal accomplishment, and depersonalization. There were significant differences in relation to satisfaction with support.
Grant et al. (2010) [57]. Developmental coaching for high school teachers: Executive coaching goes to school.	Australia: *n* = 44; Female = 70.5%; Mean age = 43.21%.	Quantitative	Secondary	Cognitive Hardiness Scale (CHS); Goal Attainment Scaling; The Depression Anxiety and Stress Scale (DASS); Workplace Wellbeing Index (WWBI); The Leadership Styles Index (LSI) (Lafferty, 1989)	The GROW model	High school teachers, educational settings in general, and corporate environments could benefit from coaching as an effective developmental intervention.
Hilger et al. (2021) [58]. Exceptional circumstances: changes in teachers’ work characteristics and well-being during COVID-19 lockdown.	Germany: *n* = 207; Female = 85%; Mean age = 34.05 years; Mean teaching experience = 6.29 years.	Quantitative	Primary (32.9%); Secondary (37.7%); Special education (13%); Vocational (16.4%).	Copenhagen Psychosocial Questionnaire; Work Fatigue Inventory; employer and work tasks; the Freiburg Bodily Complaints Inventory (FBL)	Job Demands–Resources Model (JD–R)	Decreased job demands and resources, as well as fatigue. Low job demands resulted in low fatigue and psychosomatic complaints, whereas low job resources resulted in reduced job satisfaction. Teachers with caretaking responsibilities and more experienced teachers were more vulnerable to the crisis.
Jakubowski and Sitko-Dominik (2021) [59]. Teachers’ mental health during the first two waves of the COVID-19 pandemic in Poland	Poland: *n* = 285; **Stage 1:** *n* = 145 Female = 89.7%; Mean age = 43.76 years; Mean experience = 19.07 years. **Stage 2:** *n* = 140 Female = 86.4%; Mean age = 43.76 years; Mean teaching experience = 19.07 years.	Quantitative	Stage 1: Primary (100%) Stage 2: Primary (67.86%); Secondary (32.14%).	The Depression Anxiety and Stress Scales (DASS); Berlin Social Support Scales (BSSS); the Relationship Satisfaction Scale (RS); the Injustice Experience Questionnaire (IEQ).	None	Teachers experienced mild levels of anxiety, stress, and depression during the first and second waves of the COVID-19 pandemic. Combining work and family life affected and worsened their wellbeing. Isolation also contributed to changes in social relationships and support, thereby negatively impacting their ability to effectively cope with the crisis.
Kim et al. (2022) [60]. My brain feels like a browser with 100 tabs open: A longitudinal study of teachers’ mental health and well-being during the COVID-19 pandemic.	United Kingdom: *n* = 24; Female = 75%; Mean teaching experience = 12.55 years.	Qualitative	Primary (45.8%); Secondary (54.2%).	Interview	Job Demands–Resources Model (JD–R)	The pandemic generally had a negative impact on teachers’ mental health and wellbeing, but to a lesser extent for primary school teachers.
Kittel and Leynen (2003) [61]. A study of work stressors and wellness/health outcome among Belgian school teachers	Belgium: *n* = 128; Female = 62%; Mean age = 44.4 years; Mean teaching = 19.9 years.	Quantitative	Secondary	LAKS-DOC Job Conditions; Maslach Burnout Inventory; Intensive Collaboration Questionnaire (French version)	None	Belgian teachers reported higher physical exertion, job demands, and somatic complaints, but lower job control, social support, and personal accomplishment than teachers from other European countries.
Lawrence et al. (2019) [62]. Understanding the relationship between work intensification and burnout in secondary teachers.	Australia: *n* = 215; Female = 66.5%; Mean age = 47.55 years; Mean teaching experience = 17.97 years.	Quantitative	Secondary	Intensification of Job Demands Scale (IDS); Satisfaction with Workload; Work intensification Norwegian Teacher Self-Efficacy Scale (NTSES); the Survey of Perceived Organizational Support; the Maslach Burnout Inventory—Educators Survey (MBI-ES).	Job Demands–Resources Model (JD–R)	Perceived organisational support moderated the relationship between work intensification and emotional exhaustion components of burnout.
Nie et al. (2015) [63]. The importance of autonomy support and the mediating role of work motivation for well-being: Testing self-determination theory in a Chinese work organisation.	China: *n* = 266; Female = 70.7%; Mean age = 33 years.	Quantitative	Secondary	Perceived work autonomy support scale; Teacher Work Motivation Scale; Work-Related Wellbeing Scale	None	Work motivation played an important role in teacher wellbeing. Lack of motivation predicted illness symptoms, while perceived autonomy support resulted into job satisfaction.
Ortan et al. (2021) [64]. Self-Efficacy, Job satisfaction and teacher well-being in the K-12 educational system.	Romania: *n* = 658; Female = 10.3%; Age = >20years; Teaching experience = 0–>20 years.	Quantitative	Preschool (4.4%); Primary (43.3%); Secondary (lower 26.4%, higher 25.8%).	Teachers’ Satisfaction at Work Questionnaire; Working conditions questionnaire; Relationship with Students/Colleagues Questionnaire	Structural Equation Modelling	Factors such as self-efficacy, positive student behaviour, promotion, and working conditions had significant influences on job satisfaction and wellbeing of teachers. An efficient work environment decreased attrition, burnout and emotional exhaustion.
Parrello et al. (2019) [65]. School burnout, relational, and organizational factors	Italy and Switzerland: *n* = 1068; Swiss = 964: Female = 79.7%; Mean age = 42.79 years. Italian = 104: Female = 98%; Mean age = 53.89 years.	Quantitative	Swiss: Primary (73.7%); Pre-primary (26%). Italian: Primary (80%); Kindergarten (20%).	Work Burnout Scale; The Copenhagen Burnout Inventory (CBI); Satisfaction with Life Scale (SWLS); Life Orientation Test; Organizational Identification Scale; Colleague Support Scale	None	There were no significant differences between Italian and Swiss teachers with regards to workload and burnout. However, support from colleagues had an impact on burnout in Swiss teachers, while organisational identification had a stronger impact on burnout in Italian teachers.
Peele and Wolf (2021) [66]. Depressive and anxiety symptoms in early childhood education teachers: Relations to professional well-being and absenteeism	Ghana: *n* = 444; Female = 96%; Mean age = 37 years; Mean teaching experience = 6.5 years.	Quantitative	Kindergarten	The Goldberg Anxiety and Depression Questionnaires; the emotional exhaustion domain of the Maslach Burnout Inventory.	None	High anxiety and depressive symptoms resulted in low job motivation, reduced job satisfaction, and higher levels of emotional exhaustion at the end of the school year, which led to absenteeism over the course of the school year.
Rasku and Kinnunen (2003) [67]. Job conditions and wellness among Finnish upper secondary school teachers.	Finland: *n* = 232; Female = 69%; Mean age = 46 years; Mean teaching experience = 18.4 years.	Quantitative	Upper secondary	The Leiden Quality of Work Questionnaire for Teachers (LAKS-DOC), Coping Inventory for Stressful Situations; Maslach Burnout Inventory for Educators (MBI-ES);	None	Job demands had more major effects on wellbeing than job control. High demands resulted in low job satisfaction. Job conditions and coping strategies increased the variance of somatic complaints, emotional exhaustion, and personal accomplishment.
Roeser et al. (2022) [68]. Mindfulness training improves middle school teachers’ occupational health, well-being, and interactions with students in their most stressful classrooms.	United States: *n* = 58; Female = 69%; Mean age = 41 years; Mean teaching experience = 10 years.	Quantitative	Secondary (Grades 6–8)	the Five Facet Mindfulness Questionnaire (FFMQ); Classroom Assessment Scoring System—Secondary (CLASS-S);	None	Mindfulness training for school teachers was acceptable and effective. Both teacher and classroom benefits of mindfulness training accrue over time.
Simbula (2010) [69]. Daily fluctuations in teachers’ well-being: a diary study using the Job Demands-Resources model	Italy: *n* = 61; Female = 88.5%.	Quantitative	Primary (33%); Lower secondary (27%); Upper secondary (40%).	Subscale of the Maslach Burnout Inventory (MBI)—Educator Survey, Italian version; Utrecht Work Engagement Scale (UWES), Italian version: General Health Questionnaire-12, Italian version; Job satisfaction; Teaching-related emotions	Job Demands–Resources Model (JD–R)	Teachers who could not effectively manage their work and family lives had more health impairments. Social support was beneficial to teachers’ wellbeing and job satisfaction.
Smetackova et al. (2019) [70]. Teachers between job satisfaction and burnout syndrome: What makes difference in Czech elementary schools.	Czech Republic: *n* = 2394; Female = 85%; Mean teaching = 21.42 years.	Quantitative	Primary	The Maslach Burnout Inventory—Educators Survey (MBI-ES); The Emotional Competence Questionnaire (ECQ); Social Support Questionnaire	None	Burnout was lower when teachers had strong self-efficacy, used positive coping strategy and felt satisfied.
Tadić et al. (2015) [71]. Challenge versus hindrance job demands and wellbeing: A diary study on the moderating role of job resources	Croatia: *n* = 158; Female = 82.3%; Mean age = 41.08 years; Mean teaching experience = 15.4 years.	Quantitative	Primary	Daily work-related wellbeing; Positive and Negative Affect Schedule (PANAS); Utrecht Work Engagement Scale (UWES); daily challenge and hindrance demands; Daily Job Resources Scale; Psychological Capital Questionnaire (PCQ)	Job Demands–Resources Model (JD–R)	Availability of resources helped teachers to cope with their challenging job demands.
Ullrich et al. (2012) [72]. Relationship of German elementary teachers’ occupational experience, stress, and coping resources to burnout symptoms.	Germany: *n* = 469; Female = 83.9%; Mean age = 44.45 years; Mean teaching experience = 17.63 years.	Quantitative	Primary	Classroom Appraisal of Resources and Demands (CARD); the Self-Acceptance (SAC) scale; the Maslach Burnout Inventory (MBI), German version	None	Factors affecting teacher wellbeing occurred at the individual teacher level of perceptions and varied from one school to the other. Teachers with highly demanding classrooms and low-level coping strategies experienced higher levels of burnout symptoms.
Van Horn et al. (2004) [73]. The structure of occupational well-being: A study among Dutch teachers.	Netherlands: *n* = 1252; Female = 51%; Mean age = 45 years; Mean teaching experience = 19 years.	Quantitative	Primary (59%); Secondary (28%); Vocational (13%).	Affective Wellbeing Scale; Social Wellbeing Scale; Depersonalization Scale from the MBI-ES; Cognitive Weariness Scale; psychosomatic health complaints; Organization Commitment Questionnaire; emotional exhaustion; Maslach Burnout Educator Survey (MBI-NL-ES)	A multidimensional (Warr, 1994 and Ryff, 1989) model for occupational wellbeing.	Lower wellbeing manifested in many aspects, ranging from exhaustion and lower work commitment to lack of concentration and psychosomatic complaints.
Walter and Fox (2021) [74]. Understanding teacher well-being during the COVID-19 pandemic over time: A qualitative longitudinal study.	United States: *n* = 25; Female = 96%; Age = >20 years; Teaching experience = 0–>10 years.	Qualitative	Primary and Secondary Grades: K–2nd (48%); 3rd–5th (20%); 6th–8th (16%); 9th–12th (16%).	Interview	Job Demands–Resources Model of wellbeing and the Hierarchy of Needs Theory of Motivation.	Teachers demonstrated great resilience and found effective ways to manage their wellbeing. However, individual- and managerial-level stressors contributed to burnout and attrition. Teachers felt undervalued because their voices were not incorporated into major decisions.

**Table 2 ijerph-20-06070-t002:** Teacher perceptions of personal capabilities, socioemotional competence, work conditions, and professional relationships.

Author and Year	Teachers’ Personal Capabilities (Resilience, Self-Efficacy, and Authenticity)	Socioemotional Competence (Training Opportunities, Emotional Competence, Supportive Relationship)	Personal Responses to Work Conditions (Burnout, Fatigue, Exhaustion, Stress, Unrealistic Expectations, Stress, Bureaucracy, Lack of Autonomy, and Exclusion from Decision Making)	Professional Relationships (Student Behaviours, Parents’ Relationships, Management Support Issues, and Challenges with Colleagues)
Chaudhuri et al. (2021) [3]	Teachers’ low self-efficacy was associated with distribution of focus and attention.	Not indicated.	Exhaustion and sense of inadequacy were reported. Causes of stress included managing disruptive student behaviours, differentiating instructions, challenges in assessment practices with new curricula and changes in work patterns.	The sense of inadequacy in managing student behaviour exhausted and stressed teachers and made them prone to detachment from students.
Wang et al. (2022) [7]	A combination of strategies was used to cope with stressful classroom encounters, including adaptive, problem-avoidant, or	Teachers significantly differed on teaching-related emotions, psychological wellbeing, and quitting intentions. Adaptive copers had the best psychological wellbeing and were unlikely to quit the profession. Problem-avoidant copers had the highest levels of problems, avoiding social support.	Social withdrawal copers reported the most maladaptive anxiety, anger, burnout and job dissatisfaction and the strongest intention to leave their current position and the teaching profession.	Highlighted need for professional development programmes that promote adaptive coping strategies to help struggling teachers deal with daily instructional challenges and be more resilient in the profession.
Sidek et al. (2020) [14]	Ability to manage classrooms efficiently and job satisfaction affected teacher wellbeing.	Positive teacher emotion was essential for the wellbeing of teachers and students and increased job satisfaction.	Student misbehaviour in the classroom and other forms of disruption in the work environment affected the wellbeing of teachers.	Highlighted need for professional development programmes targeting teachers’ ability to connect well with students.
Hakanen et al. (2006) [22]	Not indicated.	Peer support moderated the link between stress and positive affect.	High job demands (stressors) led to ill health and burnout	Intervention programs aimed at reducing teacher workload and access to resources were key mitigation measures. Availability of adequate resources helped teachers to work efficiently and achieve better educational goals.
Skinner et al. (2021) [23]	Teachers had low self-confidence; they felt that they were failing the children, themselves, and the profession.	Low emotional competence because teachers felt undervalued, unappreciated, deficient, and inadequate.	Causes of stress included organisational changes, increased workload and lack of managerial support. Pressures to improve students’ results and managing educational and diverse social issues in the classroom resulted in excessive workloads.	Difficulties with management and leadership styles, unfair judgement and criticisms, pressure to meet unrealistic expectations, and being under constant scrutiny by head teachers led to perceptions of being bullied. Unclear role expectations and decreased professional autonomy were associated with cumbersome bureaucratic processes.
Jepson and Forrest (2006) [31]	Teachers with high achievement striving experience had higher levels of perceived stress.	Not indicated.	Teachers reported moderate levels of perceived stress on average. Primary school teachers reported more perceived stress than secondary school teachers. No significant difference was found between males and females.	Highlighted need to identify and support teachers that were prone to stress in the workplace.
Carroll et al. (2021) [33]	Intrinsic traits such as resilience, emotion regulation capabilities, and sense of subjective wellbeing played a significant role in the experience of workplace stress.	Difficulty in regulating emotion led to greater perceived stress.	Being a teacher was the most important predictor of work- and student-related burnout, especially for early career and primary school teachers. Most teachers found their jobs either very or extremely stressful and were seriously considering leaving. Manageable workload critical.	There was need for stress reduction intervention programs that focused on workload management, effective emotional regulation, and improved sense of subjective wellbeing.
Stapleton et al. (2020) [38]	Students’ failure affected teachers’ self-efficacy because teachers tend to spend more time with their students than other professions do.	Psychological competences protected teachers from the risk of burnout.	Emotional intensity and dissatisfaction with available support predicted teachers’ levels of emotional exhaustion, depersonalization, and personal accomplishment. Male teachers experienced more burnout than their female counterparts.	Highlighted need for enhanced quality of appropriate support for teachers to reduce the risk of burnout.
Turner et al. (2022) [39]	Improvement in pedagogical practices and social support provided by colleagues led to positive wellbeing.	Teachers felt securely connected with colleagues when cared for.	Reduction in workload as a result of support from colleagues.	Support from colleagues enhanced teachers’ perceptions of being cared for.
Bermejo-Toro et al. (2016) [41]	Self-efficacy and personal coping skills were critical in modulating the effect of demands on wellbeing and played a more important role in teacher wellbeing than job resources.	Social support improved teacher wellbeing.	Teacher engagement did not directly reduce burnout levels, but it had a preventive effect.	Social support and feedback were important factors for professional wellbeing.
Burić et al. (2020) [42]	Participants with high levels of teacher self-efficacy (TSE) provided higher quality instruction and had greater power in promoting students’ motivational, affective, and cognitive outcomes.	Positive emotions were enhanced by moderate to high levels of TSE while negative emotions deteriorated TSE.	Feelings of exhaustion due to teaching.	Feelings of exhaustion due to intense interactions with students.
Kilgallon et al. (2008) [43]	Self-efficacy contributed to teacher sustenance, reflective practice, professional autonomy and contributed to job satisfaction.	Relationships developed with work colleagues and professional peers were crucial to teacher sustenance.	Teachers struggled to maintain a balance between meeting personal commitments and professional responsibilities. Reflective thinking facilitated the identification of student needs, developing realistic expectations, and modifying teaching practices.	Organisational strategies were vital to teacher access to support from line managers, parents, professional resources, education assistants, families, and friends.
Huang et al. (2019) [44]	Self-efficacy was promoted by trust in colleagues and enhanced teacher wellbeing, while emotional job demands enhanced rather than reduced participants’ self-efficacy.	Active response to emotional job demands developed better teacher–student and between-colleague relationships.	Self-monitoring was associated with anxiety and depression.	Schools needed to provide opportunities for teachers to learn more about emotional job demands and associated impacts on teaching for a more constructive and informative understanding of how to deal with such demands.
Chan et al. (2021) [45]	Teachers with high teaching efficacy felt emotionally exhausted when they were unclear about their role.	Interacting and chatting with students was motivating. Emotional support from administrators, colleagues, parents, and students was an important source of emotional stability during the COVID-19 pandemic.	There was need to reduce teachers’ workload to help reduce stress and emotional exhaustion. Teaching autonomy, social support, competency, and more flexibility were very important.	Training and resources were required to improve instructional and technological skills for distance learning. Collaboration with administrators helped to manage workload and other challenges.
Kennedy et al. (2021) [46]	Teachers’ self-efficacy for student engagement, instructional strategies, and classroom management significantly improved after six months of training.	Training enabled teachers to deal with disruptive student behaviour more effectively and to maintain a more positive classroom environment. Student wellbeing and social, emotional, and academic functioning benefitted from such teacher training.	Teacher training reduced perceived emotional stress, exhaustion, improved psychological wellbeing, and reduced burnout.	Highlighted need for organisations to implement programs that assisted primary school teachers to improve their wellbeing.
Schwarzer and Hallum (2008) [48]	Self-efficacy was a protective coping mechanism against job-related stress and burnout. Low self-efficacy led to job stress and subsequent burnout.	Not indicated.	Job stress directly led to burnout.	Highlighted need for management to invest in programs that strengthen teachers’ optimism, self-belief, and teaching skills.
Burić, et al. (2021) [53]	Deep acting (desirable positive emotions) combined with low levels of surface acting (hiding feelings and faking emotions) were related to greater professional wellbeing.	Teachers engaged in faking emotions (happiness, or enthusiasm) to promote students’ learning.	Not indicated.	Training was required to re-appraise strategies that modify negative emotional experiences and prevent overreliance on faking desirable emotions while hiding undesirable emotional turmoil.
Buric et al. (2019) [54]	Teachers with high level of perceived resilience showed lower levels of negative emotions, burnout, and psychopathological symptoms.	Burnout predicted negative emotions.	Elevated levels of burnout gave rise to psychopathological symptoms. Burnout also predicted a deterioration in mental health.	The ability to bounce back from stress (coping strategies) could be changed and modified using interventions.
Collie et al. (2016) [55]	Efficacy and autonomy were related to teacher wellbeing.	Teacher–student relationship was positively associated with teachers’ well-being among elementary school teachers.	Teacher job satisfaction increased as their competence, identified regulation, and wellbeing increased. Job satisfaction and commitment were related to wellbeing and motivation.	When the work environment was autonomous and supportive, teachers felt that their basic psychological needs were being met. Relationships with other colleagues assisted with teachers’ commitment.
Fiorilli et al. (2015) [56]	Self-efficacy was affected by students’ failure, which in turn led to increased distress and higher levels of depersonalization. Coping strategy was helpful when teachers experienced unsatisfactory support.	Emotional intensity and dissatisfaction with level of support received predicted levels of emotional exhaustion, depersonalization, and personal accomplishment. Psychological competences seemed to protect teachers from the risk of burnout.	Teachers’ wellbeing was affected by psychological variables and differences were found between Italian and Swiss teachers. Male teachers reported higher burnout levels than female participants. However, age did not have any significant effect on burnout.	Satisfactory internal and external support was required to boost teachers’ psychological competence, especially for Italian teachers.
Grant et al. (2010) [57]	The coaching group reported increased resilience and improved workplace wellbeing.	Not indicated.	Coaching reduced stress.	Participation in the coaching program was associated with enhanced leadership and communication styles.
Hilger et al. (2021) [58]	Teachers with more advanced ICT skills were better able to handle the sudden remote teaching demands compared to those without such skills.	Teachers’ direct interaction with students during face-to-face class sessions was a major motivator and source of job satisfaction.	Transition to remote teaching was just as much of a relief as it was a strain for teachers. Reduced job demands resulted in low psychosomatic complaints and fatigue.	Need for more resources and regular workshops on handling digital tools imperative for teachers. Additional support for those with caretaking responsibilities also required.
Jakubowski and Sitko-Dominik (2021) [59]	Not indicated.	Satisfactory maintenance of social support network of extreme importance.	Teachers experienced at least mild levels of stress, anxiety, and depression during the first and second waves of the COVID-19 pandemic.	Counselling and trainings on problem solving and emotional coping strategies were deemed necessary.
Kim et al. (2022) [60]	Teachers used new and existing coping strategies such as exercise and meditation to maintain their mental health and wellbeing during the COVID-19 pandemic.	Teachers found the lack of social contact with colleagues and students difficult. Social support was the strongest positive contributor to teachers’ MHWB.	Teachers’ workload increased over time and affected their MHWB. Primary school teachers were concerned about competing demands for time. Work autonomy had a positive impact on teachers’ mental health and wellbeing.	Accessibility to sources of social support and collaborative communication was deemed necessary.
Kittel and Leynen (2003) [61]	Depersonalisation was a form of defensive coping strategy against lack of personal accomplishment. Higher participation at work had a positive effect on mastery over self-efficacy and self-appraisal.	Psychological job demands were associated with burnout, especially emotional exhaustion.	Stress at work led to increased levels of burnout. Insufficient emotional resources led to depersonalisation. Males scored higher on depersonalisation but there was no difference in health outcomes associated with age or gender differences.	Higher participation at work had a positive effect on mastery over the work environment.
Lawrence et al. (2019) [62]	Not indicated.	Teachers experienced work intensification and emotional exhaustion.	Teachers were dissatisfied with nonteaching-related workload and were moderately satisfied with teaching-related workload.	Organisational support for staff was required to reduce burnout and dissatisfaction with the profession, and the risk of losing good teachers to overwork and stress.
Nie et al. (2015) [63]	Work motivation played an important role in teachers’ wellbeing.	More autonomous forms of motivation were beneficial.	Lack of motivation was related to higher work stress and increased symptoms of illness.	Organisational support for autonomy made a difference in teachers’ motivation and wellbeing.
Ortan et al. (2021) [64]	Self-efficacy and promotion had the greatest influence on job satisfaction and teacher wellbeing.	Supportive relationships with students and colleagues positively influenced job satisfaction.	Working conditions and promotion positively influenced job satisfaction and reduced teacher attrition. However, higher stress and strain from workload led to demotivation.	Relationship with management and availability of resources had significant positive influence on job satisfaction.
Parrello et al. (2019) [65]	Italian teachers showed higher optimism and identification with their schools than their Swiss colleagues.	Age and having children had impact on satisfaction with life among Swiss teachers.	There was no difference in burnout levels, support, or workload between Swiss and Italian teachers.	Organisational identification played a role in predicting burnout only among Italian teachers. On the other hand, Swiss teachers relied more on social support than Italian teachers.
Peele and Wolf (2021) [66]	Not indicated.	Not indicated.	Mental health symptoms such as anxiety and depressive symptoms increased burnout and decreased motivation and satisfaction, leading to higher absenteeism.	Workplace conditions affected mental health.
Rasku and Kinnunen (2003) [67]	Coping made a significant contribution to somatic complaints, emotional exhaustion, and personal accomplishment. Task-oriented coping was associated with high personal accomplishment.	Emotion-oriented coping strategies seemed to be linked to high levels of somatic complaints and emotional exhaustion.	Teachers had high job satisfaction and a low somatic complaint. High job demands decreased the level of wellbeing. Older participants reported higher somatic complaints than their younger colleagues. Number of work hours was not significantly associated with stress or any wellbeing indicator.	School management promoted teachers’ wellbeing by minimising time pressures and narrowing the breadth of required tasks.
Roeser et al. (2022) [68]	Participants in mindfulness training programs had better coping skills.	There was no change in teachers’ emotionally supportive interactions with students due to MT.	Teachers who engaged in the Mindfulness-Based Emotional Balance program (MBEB) reported less emotional exhaustion, anxiety, depression, and job-related stress.	Participants in mindfulness training programs showed better classroom organisation; had improved student behaviour management, with clarity about expected classroom behaviour; and spent less time on dealing with behavioural issues.
Simbula (2010) [69]	Participants’ positive mood resulted in more cooperative behaviour and students’ better task performance.	Social support was an important job resource and was related to work engagement, job satisfaction, and mental health.	Teachers who were unable to effectively manage their professional and family roles were more exhausted, which negatively affected their job satisfaction and mental health.	Highlighted need for job redesignation strategies that enhance work-related psychological wellbeing.
Smetackova et al. (2019) [70]	Self-efficacy and coping mechanisms were linked to physical, emotional, and cognitive dimensions of burnout, with high self-efficacy resulting in low emotional burnout symptoms. However, teachers had low self-efficacy in pedagogical approach.	Social support in the workplace had positive effects on job satisfaction and prevention of burnout syndrome.	On average, mild burnout was experienced by both male and female teachers. Male teachers experienced stronger emotional burnout, while female teachers experienced stronger physical burnout. Higher job satisfaction led to lower burnout syndrome and stronger self-efficacy. Negative coping resulted in burnout.	Highlighted need for high job satisfaction, workplace social support, high self-efficacy, collaboration with parents, and professional development.
Tadić et al. (2015) [71]	Not indicated.	Job resources fostered good connection with colleagues and wellbeing.	High workload to accomplish work goals was stressful due to extra demand of time and effort.	Job demands accompanied by abundant resources, including feedback and social support from colleagues, fostered positive teacher perceptions of being valued, appreciated, and supported.
Ullrich et al. (2012) [72]	Teachers with low levels of coping strategies experienced more burnout.	Not indicated.	More experienced teachers had comparatively lower burnout symptoms than their less experienced counterparts.	Not indicated.
Van Horn et al. (2004) [73]	Professional competence and autonomy were important to teachers’ self-efficacy and occupational wellbeing.	A positive social relationship with students and colleagues fostered teacher occupational wellbeing.	At the core of teachers’ occupational wellbeing were professional, social and affective dimensions of wellbeing. Lower teacher wellbeing resulted in exhaustion and lack of commitment.	Highlighted need for intervention programmes that are designed to improve teachers’ wellbeing.
Walter and Fox (2021) [74]	Teachers felt that they were unsuccessful, lacked autonomy, were inflexible, and experienced reduced efficacy. However, they demonstrated great resilience and used coping strategies to maintain their overall wellbeing during the COVID-19 pandemic.	Socioemotional wellbeing decreased since teachers preferred face-to-face teaching and felt that their ability to teach effectively had decreased.	Teachers were anxious, increasingly stressed, uncertain, and overwhelmed. Physical and mental wellbeing decreased. There was concern over the reduced impact on students’ learning.	Highlighted need for individually targeted support and school-wide strategies that emphasise care, collaboration, and opportunities for teachers’ inputs.

**Table 3 ijerph-20-06070-t003:** Quality appraisal of the reviewed studies.

Author & Year	1	2	3	4	5	6	7	8	9	10	11	12	13	14	15	16	Sum of score	% Score
Chaudhuri et al. (2021) [3]	0	2	3	2	1	2	2	2	2	2	n/a	2	2	n/a	2	2	26/42	61%
Wang et al. (2022) [7]	0	1	2	2	2	3	2	3	2	2	n/a	2	2	n/a	0	2	25/42	59.5%
Sidek et al. (2020) [14]	0	2	2	2	2	2	2	1	3	2	n/a	2	2	n/a	0	0	22/42	52.4%
Hakanen et al. (2006) [22]	3	3	2	2	3	2	2	2	3	2	n/a	2	3	n/a	0	2	31/42	73.8%
Skinner et al. (2021) [23]	0	2	2	1	2	3	2	2	n/a	n/a	2	n/a	2	2	2	1	23/39	59%
Jepson and Forrest (2006) [31]	0	2	2	2	2	3	2	2	3	2	n/a	2	1	n/a	2	0	25/42	59.5%
Carroll et al. (2022) [33]	0	3	3	2	3	3	2	3	3	2	n/a	2	1	n/a	0	2	28/42	66.7%
Stapleton et al. (2020) [38]	0	3	2	3	2	2	2	2	3	2	n/a	2	0	n/a	0	2	25/42	59.5%
Turner et al. (2022) [39]	3	2	2	2	1	2	2	2	n/a	n/a	2	n/a	2	2	2	1	25//39	64%
Bermejo-Toro et al. (2016) [41]	3	2	2	1	2	2	2	2	2	2	n/a	2	2	n/a	0	2	26/42	61%
Burić et al. (2020) [42]	3	3	3	2	3	3	2	3	3	2	n/a	2	3	n/a	0	2	34/42	81%
Kilgallon et al. (2008) [43]	0	2	3	2	2	2	3	2	n/a	n/a	2	n/a	0	0	3	1	22/39	56%
Huang et al. (2019) [44]	3	3	2	2	3	2	2	2	3	2	n/a	2	2	n/a	0	2	31/42	73.8%
Chan et al. (2021) [45]	3	2	3	2	2	3	3	3	3	3	2	2	3	2	0	3	39/48	81.3%
Kennedy et al. (2021) [46]	2	3	3	3	2	3	2	3	3	2	n/a	2	2	n/a	2	2	34/42	81%
Schwarzer and Hallum (2008) [48]	0	3	2	2	2	2	2	2	3	2	n/a	2	2	n/a	2	2	28/42	66.7%
Burić et al. (2021) [53]	0	3	3	2	3	3	2	3	3	2	n/a	2	3	n/a	0	2	31/42	73.8%
Burić et al. (2019) [[54]	0	3	3	2	3	3	2	3	3	2	n/a	2	2	n/a	0	2	30/42	71.4%
Collie et al. (2016) [55]	3	3	3	2	2	3	2	2	3	2	n/a	2	3	n/a	0	2	32/42	76.%
Fiorilli et al. (2015) [56]	0	3	2	2	2	2	2	2	3	2	n/a	2	1	n/a	0	2	25/42	59.5%
Grant et al. (2010) [57]	3	3	2	1	1	2	2	2	2	2	n/a	2	1	n/a	2	2	27/42	64.2%
Hilger et al. (2021) [58]	3	2	2	2	2	2	2	3	0	2	n/a	2	1	n/a	0	2	25/42	59.5%
Jakubowski and Sitko-Dominik (2021) [59]	0	2	2	2	2	3	2	2	2	2	n/a	2	0	0	0	2	23/42	54.3%
Kim et al. (2022) [60]	2	3	2	2	1	2	2	2	n/a	n/a	2	n/a	1	2	2	2	25/39	64.1%
Kittel and Leynen (2003) [61]	0	2	2	2	2	3	2	3	2	2	n/a	2	0	n/a	3	2	28/42	66.7%
Lawrence et al. (2019) [62]	3	3	3	2	2	3	2	3	3	2	n/a	2	1	n/a	0	2	31/42	73.8%
Nie et al. (2015) [63]	0	1	1	2	2	2	2	1	3	2	n/a	2	2	n/a	0	3	23/42	54.8%
Ortan et al. (2021) [64]	3	3	3	2	2	2	2	3	3	2	n/a	2	3	n/a	2	2	34/42	81%
Parrello et al. (2019) [65]	0	3	2	2	3	2	3	2	0	2	n/a	2	0	n/a	0	2	23/42	54.8%
Peele and Wolf (2021) [66]	0	2	3	2	2	3	2	2	2	2	n/a	2	2	n/a	2	2	28/42	66.7%
Rasku and Kinnunen (2003) [67]	0	2	2	2	2	3	2	2	3	2	n/a	2	2	n/a	0	2	27/42	64.2%
Roeser et al. (2022) [68]	0		3	2	1	3	2	2	2	2	n/a	2	2	2	1	2	26/42	61%
Simbula (2010) [69]	3	3	3	1	1	2	2	2	3	2	n/a	2	1	n/a	0	3	28/42	66.7%
Smetackova et al. (2019) [70]	0	2	2	1	2	2	2	2	3	2	n/a	2	1	n/a	0	2	23/42	54.8%
Tadić et al. (2015) [71]	3	2	2	2	2	2	2	2	2	2	n/a	2	2	n/a	0	2	27/42	64.2%
Ullrich et al. (2012) [72]	0	3	2	2	2	2	2	3	3	2	n/a	2	2	n/a	2	2	29/42	69%
Van Horn et al. (2004) [73]	3	2	2	3	3	3	2	3	2	n/a	2	2	1	n/a	1	2	31/42	73.8%
Walter & Fox (2021) [74]	3	2	1	2	2	2	2	2	n/a	n/a	2	n/a	2	2	0	2	24/39	62%

Quality Assessment Tool for Studies with Diverse Designs (QATSDD). Criteria: (1) explicit theoretical framework; (2) statement of aims/objectives in main report; (3) description of research setting; (4) sample size considered in terms of analysis; (5) representative sample of target group of a reasonable size; (6) description of procedure for data collection; (7) rationale for choice of data collection tool(s); (8) detailed recruitment data; (9) statistical assessment of reliability and validity of measurement tool(s) (Quantitative studies only); (10) fit between research question and method of data collection (Quantitative studies only); (11) fit between research question and format and content of data collection tool (Qualitative studies only); (12) fit between research question and method of analysis (Quantitative studies only); (13) good justification for analytical method selected; (14) assessment of reliability of analytical process (Qualitative studies only); (15) evidence of user involvement in design; (16) strengths and limitations critically discussed. Indicator: 0 = not at all; 1 = less likely; 2 = fairly likely; 3 = completely likely; n/a = not applicable.

## Data Availability

Data sharing is not applicable to this article.

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
