# Peer review of "A Systematic Review of the Factors That Influence Teachers’ Occupational Wellbeing"

_ijerph, 2023, doi:10.3390/ijerph20126070_

Round 1

Reviewer 1 Report

The presentation of the results should be better synthetised. Currently, it is somehow difficult to follow and not enough structured. For example:

·       A definition of terms used in this cathegorization would be useful; otherwise why should the authors separate “the personal teacher capabilities” from the socio-emotional competence” which is also included by others in broader definition “personal capability”?

·       The paragraph 3.3.1. Personal teacher capabilities should be adjusted, as it

combines personal characteristics such as “high self efficacy with work stressors such as lack of autonomy and excessive bureaucracy. The last ones shoud be included in the paragraph referring to the working conditions, as they mostly depend on the processes inside the organization are defined.

·       Line 237 – it is very unclear what influences what: the perceived self efficacy influences the positive mood or viceversa? The description of this relation should be presented in more detail.

·       Line 240 – what is the exactly meaning of “decrease in socio-emotional competence” reported during COVID-19? It should be specified, not mentioned in a generic way.

·       Line 252-254: The following phrase should be reformulated:  “Teachers’ personal responses to work conditions, workload, burnout, exhaustion, stress, and fatigue were discussed in 25 studies”, as burnout, exhaustion and fatigue are responses to work conditions and workload. The responses cannot be included in the same list with their causes, as correctly are mentioned in the following lines.

·       Line 259-260: The following phrase is also difficult to understand: “Teachers also reported stress due to anxiety, depression, and unrealistic expectations during remote teaching”. Do you mean that during remote teaching, there were unrealistic expectations (please mention them) which created stress? Or they created anxiety and depression? Or remote teaching created anxiety that led to stress? Or to some teachers was stresfull, to other creted anxiety and depression? The phrase should be more specific.

·       Line 274 and line 308 describe factors related to job satisfaction. It would be easier to include in one paragraph everything related to job satisfaction; particulalry if talking about job resources which is another element of the working condition. The same for the “positive moods”, which should be linked to the previous references about this topic.

The discussion part is reproducing the results and should be also more synthetic. The part “implications for practice” should be sufficient for the discussion section.

Reviewer 2 Report

Could the tables be a little shorter?

Reviewer 3 Report

I would like to congratulate authors for this relevant piece of work. This systematic review is sorely needed in order to determine what the next steps are for promoting teacher wellbeing. The research design is rigorous and fulfills the required criteria. So I will make only two suggestions:

1. The ownership of schools, whether they are public, charter, or state  schools, has not been added. The subject taught by teachers has not been considered either. The analysis of these variables could have enriched the results of this study. Therefore, this should be addressed in the study limitations. I believe that this part has not been sufficiently developed (and strengths have been prioritized over the limitations).

2. Future lines of research arising from this study should be more clearly indicated (and developed if possible).

Reviewer 4 Report

Needs to address (limitations and future direction) in more detailed way. Both Table 2 and 3 need more elaborations. The COVID is mentioned briefly. Some elaborations needed as the effects of more online teaching on teachers. The tables carry many words (N/A), what is the effect of (not available) of the credibility of the overall conclusions. Singapore usually ranks top in education achievement; however, I did not see it listed. In the discussion part, there needs to be more elaborations on the (countries) and teacher's wellbeing. Needs also to elaborate on the (advantage) of the method used compared to direct in-depth survey. One limitation is that I do not see elaborations on (differences) between countries. 

Reviewer 5 Report

The paper provides an interesting review of recent studies on factors affecting the occupational well-being of non-university teachers.

The process of screening and selection of articles using an appropriate database is pertinent, and the work of synthesising and compiling the main variables present in the studies analysed is equally pertinent and constitutes a valuable summary of the studies carried out to date on this important and recurrent subject. The compilation of the main variables present in the studies analysed is equally pertinent and constitutes a valuable summary of the studies carried out to date on this important and recurrent subject.

It would be advisable to list the different variables analysed on the basis of the generic dimensions that are the object of the study in a table of contents.

As a suggestion for a later study, we recommend extending the selection to studies related to bullying in the teaching environment, a subject which is also directly studied in this type of work. We also recommend that for other studies, using similar selective processes, the analysis should include university teaching staff.

Round 2

Reviewer 4 Report

I think the revised version is fine.